# A Little Robustness Goes a Long Way: Leveraging Robust Features for Targeted Transfer Attacks

**Jacob M. Springer**
Los Alamos National Laboratory
Los Alamos, NM, 87545
jacmspringer@gmail.com

**Melanie Mitchell**
Santa Fe Institute
Santa Fe, NM, 87501
mm@santafe.edu

**Garrett T. Kenyon**
Los Alamos National Laboratory
Los Alamos, NM, 87545
gkenyon@lanl.gov

## Abstract

Adversarial examples for neural network image classifiers are known to be transferable: examples optimized to be misclassified by a source classifier are often misclassified as well by classifiers with different architectures. However, *targeted* adversarial examples—optimized to be classified as a chosen target class—tend to be less transferable between architectures. While prior research on constructing transferable targeted attacks has focused on improving the optimization procedure, in this work we examine the role of the source classifier. Here, we show that training the source classifier to be "slightly robust"—that is, robust to small-magnitude adversarial examples—substantially improves the transferability of class-targeted and representation-targeted adversarial attacks, even between architectures as different as convolutional neural networks and transformers. The results we present provide insight into the nature of adversarial examples as well as the mechanisms underlying so-called "robust" classifiers.

## 1   Introduction

Neural-network image classifiers are well-known to be susceptible to adversarial examples—images that are perturbed in a way that is largely imperceptable to humans but that cause the neural network to make misclassifications. Much research has gone into methods for constructing adverarial examples in order to understand the nature of neural-network vulnerabilities and to develop methods to make neural-network classifiers more robust to attacks [7, 11, 21, 45, 50, 55, 73].

*Untargeted* adversarial examples are designed to elicit an unspecified incorrect class, while *targeted* adversarial examples are designed to elicit a specific (incorrect) *target* class. A given adversarial perturbation is designed via optimization with respect to a given trained network; here we call this the *source* network. While untargeted adversarial examples are often transferable—examples designed to attack a source network also successfully attack other trained networks with different parameters and architectures [21, 73]—targeted adversarial examples tend to be less transferable [44].

Prior research on constructing transferable adversarial examples for image classifiers has focused primarily on improving optimization methods for generating successful image perturbations. In this paper, we take a different tack—focusing on the source neural network used in constructing adversarial examples. Specifically, we find that "slightly-robust" convolutional neural networks (CNNs)—ones that have been trained to be robust to small adversarial perturbations—can be leveraged to substantially improve the transferability of targeted adversarial examples to different architectures. Surprisingly, we show that targeted adversarial examples constructed with respect to a slightly robust CNN transfer successfully not only to different CNN architectures but also to transformer architectures such as ViT [16], LeViT [22], CCT [24], and even CLIP [57], which is trained on different objective than the source CNN. Such transfers are all "black box" attacks—while generating an adversarial example

35th Conference on Neural Information Processing Systems (NeurIPS 2021).

requires knowledge of the source network's architecture, no such knowledge is required for the black-box architectures that can also be attacked by the same example.

The vulnerability of networks to both targeted and untargeted attacks has huge significance for security purposes, and thus understanding how to construct highly effective attacks can help motivate defenses. However, we believe that targeted attacks are especially important for understanding neural-network classifiers, as they provide a tool to compare the features of two models. When a targeted attack transfers from one network to another, it suggests that the two networks rely on similar information for classification, and that they use the information in the same way. In our work, we show that each individual slightly-robust neural network transfers features effectively to all tested non-robust networks, suggesting the surprising result that slightly-robust networks rely on features that overlap with every non-robust network, even though it is not the case that any particular non-robust network has features that substantially overlap with all other non-robust networks.

In addition, we leverage the techniques of adversarial transferability to examine which features are learned by neural networks. In accordance with prior work [71], we find that, on the spectrum from non-robust (standard) to highly robust classifiers, those that are only *slightly* robust exhibit the most transferable representation-targeted adversarial examples, suggesting that the features of slightly-robust networks overlap substantially with *every* tested desination network. This can explain why slightly robust networks give rise to more transferable adversarial attacks and have better weight initializations for downstream transfer-learning tasks [43, 62, 76, 80].

The main contributions of this paper are the following:

1. We demonstrate that adversarial examples generated with respect to slightly robust CNNs are more transferable than those generated with respect to standard (non-robust) networks, This transferability extends not only to other CNNs, but also to transformer architectures.
2. We find that, as the robustness of the source network increases, there is also a substantial increase in transferability of targeted adversarial examples to *adversarially-defended* networks.
3. We examine the role of the adversarial loss function in generating transferable adversarial examples.
4. We show, surprisingly, that non-robust neural networks do not exhibit substantial feature (representation) transferability, while slightly-robust neural networks do. This helps explain why slightly-robust neural networks enable superior transferability of targeted adversarial examples.

## 2   Background

**Adversarial Examples.**   In this paper we are primarily concerned with properties of source networks that facilitate transferability of adversarial examples. Let $F : \mathcal{X} \to \mathcal{Y}$ denote a "white-box" network (i.e., one whose architecture and weights are known to the adversary) and let $G : \mathcal{X} \to \mathcal{Y}$ denote a "black-box" network (weights and architecture are unknown to the adversary). Let $(x, y) \in \mathcal{X} \times \mathcal{Y}$ be an (unperturbed) input-label pair, where $\mathcal{X}$ is the input-space and $\mathcal{Y}$ is the label-space. Given a maximum perturbation size $\varepsilon$, we construct an adversarial example $x + \delta$ where $\|\delta\|_\infty \leq \varepsilon$, such that $F(x + \delta) \neq y$ for the *untargeted* case, and $F(x + \delta) = t$ for some target class $t \in \mathcal{Y}$ for the *targeted* case. We then say that $x + \delta$ is transferable to black-box network $G$ if $G(x + \delta) \neq y$ for the untargeted case and $G(x + \delta) = t$ for the targeted case.

**Optimizers.**   Prior research has identified a number of methods for optimizing adversarial examples given a white-box classifier $F$, many of which are based on the Iterative Fast Gradient Sign Method (I-FGSM) [21, 40], in which a perturbation $\delta_i$ is iteratively updated to maximize the loss of the network while obeying an $\ell_\infty$ norm constraint.

We adopt the state-of-the-art method recently proposed by [91], which combines three variants of I-FGSM and optimizes over many steps:

1. Diverse Input Iterative Fast Gradient Sign Method (DI$^2$-FGSM), which applies a random affine transformation to the input at each step prior to computing the gradient [87],
2. Translation-Invariant Iterative Fast Gradient Sign Method (TI-FGSM), which convolves the gradient with a Gaussian filter [14],
3. Momentum Iterative Fast Gradient Sign Method (MI-FGSM), in which a momentum term is added to the gradient [13].

We follow the convention of Zhao et al. [91] and call the combination of these processes *TMDI-FGSM*. We describe the method in detail in the appendix.

For targeted adversarial examples, the loss function $L$ should be maximized when the target label is predicted with high confidence. For untargeted adversarial examples, this occurs when the predicted label differs from the true label, and when the true label is given a low confidence. A number of adversarial loss functions have been proposed, including standard cross-entropy loss [73], CW loss [7], and feature-disruption loss [34]. We use the highly-effective logit loss, proposed by [91], which is maximized for targeted examples when the logit score for a target class (i.e., the value of the output neuron associated with the target class prior to the softmax operation) is maximized. Similarly, the untargeted version aims to minimize the logit score associated with the true class.

**Constructing Robust Source Networks.** We construct robust source networks by performing adversarial training with projected gradient descent [21, 45]. Each source network is trained to be robust to adversarial examples with $\ell_2$ norm less than a specified $\varepsilon$ parameter, which we call the *robustness parameter*. For this paper, we rely on pre-trained robust ImageNet models [62], which have been released under the MIT License. These source networks, along with many of our experiments, are implemented in PyTorch [56].

**Features.** In this paper, we will refer to neural network *features* [32, 71, 78]. A feature $f : \mathcal{X} \to \mathbb{R}$ maps input to a real number to describe how "strongly" the feature appears in the image. Every neuron in a neural network computes a feature. However, we are primarily concerned with the *representation-layer* features, i.e., the features computed by the neurons in the penultimate layer [32]. When we are referring to the features of a specific neural network, we are referring to the features described by the neurons in the representation layer. When we say that the features of two different neural networks *overlap*, we mean that the patterns of the input that affect the features of one neural network also affect the features of the other neural network. When features are easily manipulated by small perturbations to the input, they are said to be *non-robust*; likewise when they are not easily manipulated in this way, they are said to be *robust* [32]. Robust networks, i.e., networks that are less vulnerable to adversarial perturbations, should rely primarily on robust features, though non-robust networks can rely on a mixture of non-robust and robust features [32, 71].

## 3 Transferability of Adversarial Examples

In this section, we describe the methodology and results of our experiments on the transferability of adversarial examples as a function of the robustness of the source network. We evaluate the targeted and untargeted effectiveness of each constructed adversarial example on a collection of convolutional classifiers, Xception [9], VGG [68], ResNet [25, 26], Inception [74], MobileNet [29], DenseNet [30], NASNetLarge [93], and EfficientNet [75], as well as transformer-based classifiers, ViT [16], LeViT [22], CCT [24], and CLIP [57]. Here we use the term *destination network* to denote the networks on which we will evaluate transferability of adversarial examples that were generated with respect to a source network. For our ImageNet experiments, we rely on pre-trained models [10, 62].

**Generating Adversarial Examples.** We choose 1000 images randomly from the ImageNet validation dataset such that every image has a different class. We generate target classes randomly for each image such that each class is a target for exactly one image, and no image has a target that is the same as its true class. For each classifier (of different robustness), we generate targeted adversarial examples for each of the 1000 selected images, targeting the associated target class. To optimize each adversarial example, we run the TMDI-FGSM algorithm for 300 iterations. When the exact image input dimensions differs between the source and destination network, we rescale the adversarial example to fit the dimensions required by the destination network using bilinear interpolation. To generate adversarial examples, we use the Robustness library [17].

**Transferability to Convolutional Network Classifiers.** We find that across every destination convolutional network, using adversarial examples optimized with respect to a source network with small robustness parameter improves transfer success rate in both the targeted and untargeted setting compared to the success rate of the non-robust ($\varepsilon = 0$) network (Figure 1). Our results on untargeted adversarial examples can be found in the Appendix. The success peak is approximately the same

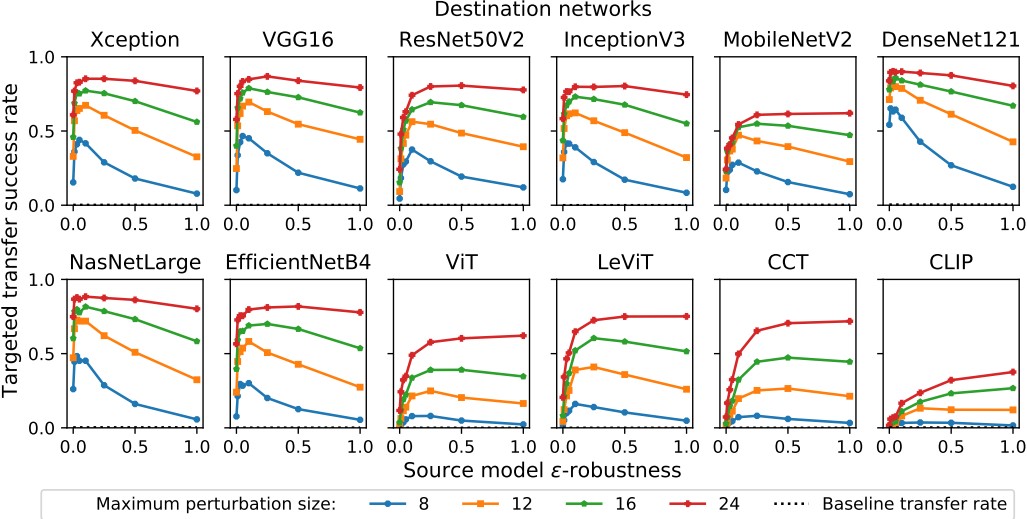

Figure 1: Targeted transfer attack success rate against ImageNet classifiers using adversarial examples optimized with respect to $\varepsilon$-robust ResNet50 source models. Success rate is the fraction of adversarial examples classified as their adversarial target by the destination network. Higher is a more successful attack. Baseline refers the rate at which unperturbed images are classified as the target class. (Best viewed in color.)

across every convolutional destination network ($\varepsilon = 0.1$), suggesting that there is an optimal source-network robustness parameter in order to maximize transferability to convolutional architectures trained on ImageNet. Note that our attacks do not exclude images that are already misclassified (untargeted case), and misclassified as the target class (targeted case) prior to adversarial perturbation. In Figure 1, we plot the baseline performance of the attacks, i.e., the performance of the attacks for unperturbed images.

Interestingly, we observe a substantial drop in attack success as robustness increases past the optimal value. While not shown in Figure 1, this trend continues as $\varepsilon$ increases above 1 (see Appendix). We hypothesize that as the robustness of a source network increases beyond a certain point, the network begins to entirely ignore many of the non-robust features relied upon by the (non-robust) destination networks, and thus attacks do not modify these features, reducing attack success. This is consistent with prior literature [78]. We dedicate the second half of this paper to describing a possible mechanism to explain these results.

**Transformer-Based Classifiers.** Few studies have addressed the robustness of transformer-based classifiers to transfer attacks [66]. To our knowledge, our paper is the first to address *targeted* transfer attacks against transformer architectures. Shao et al. [66] finds that transformer-based image classifiers are more robust to transfer attacks than convolutional classifiers, especially when the source model is convolutional. We find an even more striking result: with only minimal attack performance against transformer networks, previously published methods that use non-robust source networks are almost entirely ineffective at constructing targeted transferable adversarial examples using convolutional source models (Table 1). This suggests that the features learned by transformer-based models and *non-robust* convolutional models are largely different.

However, we find that using a *slightly-robust* ResNet50 classifier as a source network dramatically improves the transferability of targeted adversarial examples to transformer-based classifiers. The optimal robustness parameter for the source network is different for destination transformer networks and destination convolutional networks, though we find that any amount of robustness in the source model (below a critical value) substantially improves transferability. Thus, in a real-world black-box attack setting—in which the destination network's architecture is entirely unknown—an adversary could find a balance between a source network robustness parameter that optimizes transfer performance for CNN classifiers and for transformer-based classifiers.

Table 1: Direct comparison of targeted transfer attack success rate between our technique (slightly-robust source models, i.e., $\varepsilon > 0$) and previously proposed strong baseline attacks (non-robust source models, i.e., $\varepsilon = 0$). We compare three different loss functions: cross-entropy, Poincaré distance combined with triplet loss, and logit loss. In addition, we report the success rate of FDA from the original paper (see text for discussion). We limit the $\ell_\infty$ norm of the adversarial examples to the standard value of $16/255$.

|  |  | Xcept | VGG16 | RN50v2 | IncV3 | MNv2 | DN121 | NNL | ENB4 | ViT | CLIP |
|---|---|---|---|---|---|---|---|---|---|---|---|
| Xent | $\varepsilon = 0$ | 10.4 | 9.6 | 4.6 | 10.6 | 6.4 | 40.5 | 13.1 | 6.8 | 0.8 | 0.1 |
| Po+Trip | $\varepsilon = 0$ | 20.8 | 15.2 | 10.0 | 23.0 | 11.6 | 59.3 | 31.2 | 14.2 | 1.3 | 0.3 |
| Logit | $\varepsilon = 0$ | 45.9 | 40.0 | 15.3 | 43.6 | 22.9 | 77.9 | 60.3 | 39.6 | 3.9 | 0.4 |
| FDA* | $\varepsilon = 0$ | – | 43.5 | – | – | 22.9 | 57.9 | – | – | – | – |
| Xent | $\varepsilon = 0.1$ | 54.0 | 59.4 | 45.8 | 50.8 | 32.1 | 78.8 | 66.0 | 41.1 | 8.6 | 2.4 |
| Po+Trip | $\varepsilon = 0.1$ | 59.1 | 57.9 | 53.0 | 56.5 | 39.2 | 78.4 | 72.6 | 45.1 | 11.4 | 3.3 |
| Logit | $\varepsilon = 0.1$ | **77.2** | **78.8** | 64.5 | **73.1** | **52.5** | **84.0** | **81.6** | **68.9** | 33.4 | 11.2 |
| Xent | $\varepsilon = 1$ | 60.4 | 69.3 | **66.6** | 58.2 | 46.7 | 69.9 | 61.3 | 56.9 | 29.9 | 19.9 |
| Po+Trip | $\varepsilon = 1$ | 48.5 | 54.4 | 60.2 | 49.5 | 39.9 | 62.6 | 53.3 | 45.0 | 22.0 | 12.4 |
| Logit | $\varepsilon = 1$ | 56.1 | 62.4 | 59.5 | 55.0 | 47.2 | 67.0 | 58.3 | 53.6 | **36.0** | **26.7** |

**CLIP.** Radford et al. [57] describes CLIP, a transformer-based classifier based on the ViT architecture. CLIP is trained to simultaneously encode images and short textual descriptions of the images. CLIP can be used for highly effective zero-shot classification by determining which class label, encoded as text, has an encoding most similar to that of the input image. Despite the fact that CLIP has not been explicitly trained with ImageNet labels or to optimize for the classification task, we find that the transfer performance of targeted adversarial examples is improved substantially when the source network (ResNet50) is slightly robust (Table 1, rightmost column), again suggesting that slightly-robust neural networks rely on features which overlap with non-robust networks, even when the non-robust networks differ in architecture and training algorithm.

For our experiments, we use ViT-B/32, the CLIP architecture based on ViT-B/32, CCT-14t/7x2, and LeViT-256.

**Improvements Upon Existing Attacks.** We directly compare the targeted transfer attack success rate to previous state-of-the-art black-box attacks and find that our method substantially outperforms the previous methods under similar constraints (Table 1). In particular, we evaluate our method's attack performace with three different loss functions: standard cross-entropy loss (Xent), Poincaré distance with a triplet loss term (Po+Trip) [41], and logit loss. We include a comparison with the feature distribution attack (FDA) [33], however, FDA requires that we train multiple supplemental models for each individual target class, which would require thousands of supplemental models to attack all thousand classes of ImageNet. Thus, we do not perform a direct comparison and instead report the targeted transfer attack success rate that is reported in the original FDA paper [33].

**Attacking Adversarially-Trained Models.** Adversarial training has been shown to improve robustness to transfer attacks [45]. We evaluate the transferability of adversarial examples to adversarially trained destination networks. Even though the adversarial perturbations which we use to attack each adversarially-trained network are larger than the magnitude for which the destination networks are trained to be robust, the adversarial examples generated using non-robust networks do not transfer to the adversarially trained networks. However, as shown in Figure 2, as the robustness of the source network increases, the attack success rate increases substantially. Similar to Figure 1, Figure 2 includes the baseline rate at which the destination network classifies unperturbed images as the (incorrect) target class.

**Extending Our Methods to CIFAR-10.** We repeat many of our experiments for the CIFAR-10 dataset [39]. The results are consistent with our ImageNet results. We present and discuss these results in detail in the Appendix.

**Computational Limitations.** Due to the computational requirement of training multiple robust ImageNet classifiers, we restrict our experiments to those we can compute using pre-trained robust

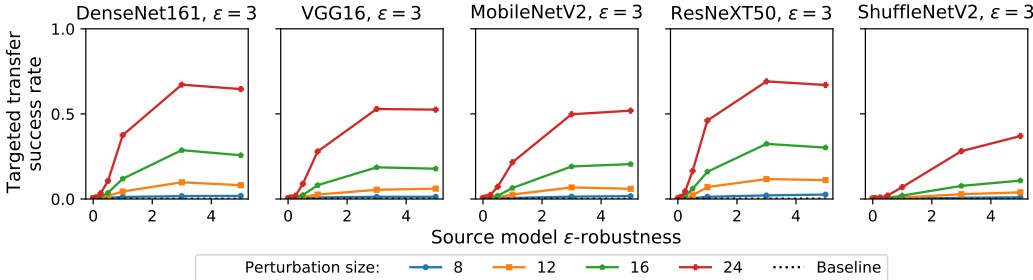

Figure 2: Targeted transfer attack success rate against adversarially trained ($\varepsilon = 3$) destination ImageNet classifiers, where examples are generated using $\varepsilon$-robust ResNet50 source networks. Higher is a more successful attack. Baseline refers the rate at which unperturbed images are classified as the target class. (Best viewed in color.)

networks. Thus, we test adversarial attacks only using the ResNet50 architecture as a source network, and we evaluate attacks only on destination networks which are easily available to us. Similarly, we do not compute ensemble attacks using slightly-robust source networks, although we expect this technique to improve the success of our attacks.

# 4    Adversarial Transferability of Features

We have shown that we can construct *class-targeted* adversarial examples that transfer a specific (incorrectly) predicted class to destination networks. In this section, we aim to address an important question to help us answer *why* class-targeted adversarial examples transfer: *to what extent do class-targeted adversarial examples transfer the representation-layer features across networks?* One could imagine that class-targeted transferability arises from the overlap of a small number of especially vulnerable features where manipulation of these features can change the model classification, or, alternatively, the overlap of many features. To answer this question, we will construct and evaluate *representation-targeted* adversarial examples. We refer to the degree to which adversarial examples of a source model can analogously affect the features that are computed by the *representation layer* of the destination model as the *representation transferability* from source to destination. By contrast, *class transferability* (what is commonly referred to as just *transferability*) refers to the degree to which adversarial examples can analogously affect the output of the destination network.

**Representation Transferability.**    Our goal is to study the representation transferability of source classifiers (with varying degrees of robustness) to non-robust models. We will show that slightly-robust networks exhibit a substantially higher degree of representation transferability than non-robust networks and more-robust networks. This can directly explain why class-targeted adversarial examples constructed using slightly-robust source networks are more transferable, as adversarial examples generated with slightly-robust networks will broadly transfer features, and will thus rely less on a small number of highly-vulnerable features that may not be present in every model. In this section, we show that representation-targeted adversarial examples generated with slightly-robust networks are highly transferable, even across a substantial difference in network architecture, such as the difference between CNNs and transformer networks.

**Measuring Representation Transferability.**    To measure representation transferabililty, we rely on a simple but powerful test to assess the similarity between the representations produced by two different networks. Let $x$ and $y$ be two inputs that produce identical (or very similar) patterns of activation in the source network's representation (penultimate) layer. If the source network has a high degree of representation transferability to the destination network, then the responses to $x$ and $y$ will be very similar in the destination network as well. By contrast, if $x$ and $y$ do not share similar representations in the destination network, then the source network has a low degree of representation transferability.

This method allows us to test the representation transferability of a source network to a destination network by constructing images $x$ and $y$ with similar representations in a source network and

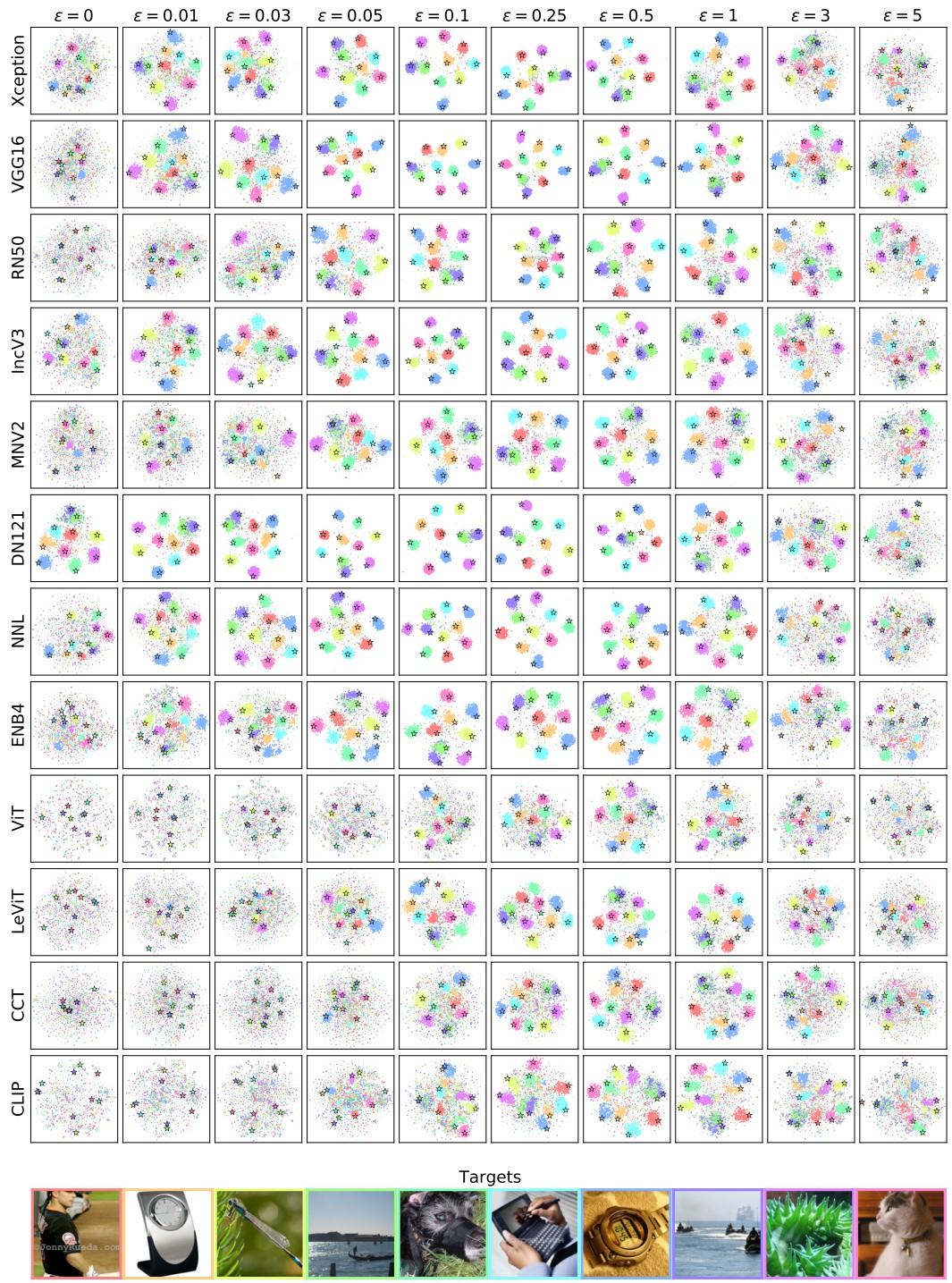

Figure 3: t-SNE plots of destination-network representations of representation-targeted adversarial examples generated by using whitebox ResNet50 models of specified $\varepsilon$-robustness. The 10 images at the bottom are the ones from ImageNet that we use as representation targets, as described in the text. (Best viewed in color and magnified.)

Table 2: Cosine similarity between representations (in the destination network) of representation-targeted adversarial examples $y$ and the corresponding target image $x$, as a function of robustness parameter $\varepsilon$ of the source network. Each value is an average over the 9900 $(x, y)$ pairs of representation vectors.

| Destination | Source network robustness parameter ($\varepsilon$) | | | | | | | | | |
|---|---|---|---|---|---|---|---|---|---|---|
| | 0 | 0.01 | 0.03 | 0.05 | 0.1 | 0.25 | 0.5 | 1 | 3 | 5 |
| Xception | 0.462 | 0.505 | 0.531 | 0.563 | **0.594** | 0.585 | 0.572 | 0.543 | 0.449 | 0.404 |
| VGG16 | 0.333 | 0.401 | 0.417 | 0.494 | **0.528** | 0.520 | 0.520 | 0.486 | 0.383 | 0.333 |
| ResNet50V2 | 0.284 | 0.348 | 0.379 | 0.432 | 0.497 | 0.496 | **0.510** | 0.484 | 0.380 | 0.321 |
| InceptionV3 | 0.577 | 0.612 | 0.627 | 0.644 | **0.673** | 0.662 | 0.655 | 0.636 | 0.572 | 0.539 |
| MobileNetV2 | 0.431 | 0.459 | 0.460 | 0.493 | **0.517** | 0.513 | 0.513 | 0.504 | 0.455 | 0.425 |
| DenseNet121 | 0.672 | 0.689 | 0.685 | 0.713 | **0.726** | 0.714 | 0.706 | 0.679 | 0.616 | 0.584 |
| NasNetLarge | 0.356 | 0.422 | 0.452 | 0.488 | **0.541** | 0.513 | 0.482 | 0.437 | 0.315 | 0.271 |
| EfficientNetB4 | 0.085 | 0.111 | 0.137 | 0.144 | **0.237** | 0.220 | 0.226 | 0.202 | 0.112 | 0.074 |
| ViT | 0.066 | 0.087 | 0.109 | 0.129 | 0.195 | **0.206** | **0.206** | 0.203 | 0.120 | 0.086 |
| LeViT | 0.051 | 0.077 | 0.096 | 0.111 | 0.165 | 0.163 | **0.176** | **0.176** | 0.107 | 0.07 |
| CCT | 0.048 | 0.081 | 0.109 | 0.137 | 0.202 | 0.227 | **0.248** | 0.241 | 0.144 | 0.093 |
| CLIP | 0.529 | 0.541 | 0.550 | 0.563 | 0.585 | 0.599 | 0.606 | **0.613** | 0.581 | 0.566 |

measuring the similarity of the representations in the destination network. To construct these images, we select two images, $x$ and $y_0$ from the ImageNet testset. We construct a representation-targeted adversarial example $y = y_0 + \delta$ targeting the representation of $x$. More precisely, we run the TMDI-FGSM algorithm to minimize the the $\ell_2$ distance between the representations of $x$ and $y$:

$$\delta = \underset{\|\delta\|_\infty \leq \varepsilon}{\arg\min} \|F^{\text{rep}}(x) - F^{\text{rep}}(y_0 + \delta)\|_2$$

where $F^{\text{rep}}$ represents the representation layer of the source network. Since we want to observe how well the representations transfer under the conditions of typical adversarial examples, we limit the perturbation to have an $\ell_\infty$ norm of $\varepsilon$, which, for this paper, we set to be the standard $16/255$. To limit the computational requirements of this experiment, we randomly select 10 images whose representations we use as targets $x$ and 990 images to use as initial images $y_0$. Of these 1000 total images, no two share the same ImageNet class. For each source network, and for each initial image $y_0$, we construct ten representation-targeted adversarial examples—one for each target $x$—for a total of 9900 representation-targeted adversarial examples per source classifier. For each target $x$, the 990 representation-targeted adversarial examples $y$ will have similar representations to $x$ in the source network. By measuring the similarity of the $(x, y)$ representations in the destination network, we can measure the representation transferability of the source network.

Here we present two different similarity metrics for comparing representations of $x$ and $y$. First, we plot the t-distributed stochastic neighbor embedding (t-SNE) of the representation vectors of each representation-targeted adversarial example $y$ in the destination network (Figure 3). Each color corresponds to one of the 10 target images ($x$). The 10 stars in each plot correspond to the t-SNE embedding of the destination-network representation of each target $x$. If the destination-network representation of each $y$ associated with a particular color is near to its corresponding star, the representation transferability is high. When representation transferability is low, the destination-network representations of representation-targeted adversarial examples that target the same image will be dissimilar, and thus not appear grouped in Figure 3. By contrast, when representation-targeted adversarial examples transfer successfully, we observe clusters grouped by target image (in Figure 3, by color). The visual tightness of each cluster represents the similarity between the representations associated with each point and its neighbors. As a second similarity metric, we report the mean cosine similarity of the representations (in destination networks) between $(x, y)$ pairs, averaged over all 9900 such pairs for each source network (Table 2). The higher the mean cosine similarity, the higher degree of representation transferability from the source network to the corresponding destination network.

**Representation Transferability Is Poor in Non-Robust Networks.** Our first surprising result is that representation-targeted adversarial examples generated with standard (i.e., non-robust) networks do not have substantial representation transferability to most tested destination networks ($\varepsilon = 0$ columns of Figure 3 and Table 2). This suggests that even when the adversarial classification output

is successfully transferred, the individual features of the destination networks are not substantially perturbed, suggesting that transferability from non-robust source networks arises from only a slight overlap in features, or the overlap of only a few highly vulnerable features. The result implies that the features of non-robust networks may not overlap substantially with each other. Interestingly, we observed some degree of representation transferability of adversarial examples generated with non-robust networks to DenseNet121, which may explain the high degree of class-transferability to DenseNet121 observed in Figure 1.

**Slightly-Robust Networks Exhibit Good Representation Transferability.** We find that representation-targeted adversarial examples generated with slightly-robust networks (approximately $0.03 \leq \varepsilon \leq 1$) have a high degree of representation transferability (i.e., clusters are tight and cosine similarity is high for these values in Figure 3 and Table 2). This representation transferability appears to peak, for convolutional networks, approximately when $0.1 \leq \varepsilon \leq 0.5$, which is coincident with the optimal source robustness for class-targeted transferability (Figure 1). For the transformer networks, including ViT and CLIP, representation transferability peaks when $0.5 \leq \varepsilon \leq 1$, which, similarly, occurs close to the optimal robustness parameter for class-targeted transferability. Surprisingly, representation transferability from slightly-robust ResNet50 classifiers to CLIP was high, despite the fact that CLIP is not trained on the traditional classification problem and is instead trained to encode images and a corresponding textual description similarly. The high degree of representation transferability of slightly-robust networks likely explains the effectiveness of slightly-robust networks for generating class-targeted adversarial examples. More broadly, the high degree of representation transferability suggests that the features of slightly-robust networks overlap substantially with the features of *every* tested (non-robust) destination network, which is the claim of Springer et al. [71].

**Representation Transferability in Very Robust Networks is Poor.** Interestingly, when networks are adversarially-trained with a large $\varepsilon$ parameter, the degree to which the features they learn overlap with the features of non-robust networks diminishes as $\varepsilon$ increases. We speculate that certain non-robust features are ignored by robust neural networks with a sufficiently large $\varepsilon$ parameter, and thus as robustness increases past a point, many of the features of non-robust networks are ignored and representation transferability decreases.

## 5   Related Work

The vulnerability and defense of neural networks to adversarial examples have been studied extensively [1, 3, 5–8, 11, 18, 19, 21, 27, 37, 45, 49, 50, 55, 59, 63, 64, 72, 73, 79, 83].

First proposed by Goodfellow et al. [21], transferable adversarial examples are generally constructed by optimizing a perturbation to fool a white-box (source) network with hopes that it will transfer to black-box (destination) networks. Recent transferability research has focused on improving the optimizer to add generalization priors [40, 54, 88, 92]: researchers have added momentum to the gradient descent process [13], introduced transformations to the input [14, 67, 87], modified the adversarial loss functions [7, 41, 91], attacked intermediate feature representations [31, 33–35, 60, 92], linearizing the source network [23], and additional manipulations to the gradient computation [85]. Additionally, some research has examined the role of source classifier(s): many of the aforementioned papers test the difference between source classifier architectures and researchers have proposed using generative networks [70, 90], and ensemble attacks [44, 77].

Robust networks have been shown to have a number of valuable properties, including serving as a good starting point for transfer learning [43, 62, 76, 80] and gradient interpretability [18]. The vulnerable features learned by neural networks have been studied both empirically [4, 15, 20, 32, 36, 46, 47, 52, 69, 71, 82, 84, 89] and theoretically [1, 2, 12, 28, 51, 65, 81, 86]. Similarly, there has been some research related to the so-called "universality" hypothesis [38, 42, 53, 58], which speculates that under the right conditions, all neural networks may learn analogous features.

In prior work, the similarity of the feature spaces of different neural networks has been compared via linear transformations; however, these techniques are either anecdotal or do not account for possible non-linear relationship between neural networks [38, 42, 53, 58].

Our research draws inspiration from recent work [71] that proposes that slightly-robust features exhibit the universality principle, and presents a limited experiment that slightly-robust networks

can be used to increase transferability. By contrast, our work provides a comprehensive study demonstrating that the technique can be used to achieve state-of-the-art transferability across a wide variety of architectures.

## 6 Conclusion

We have demonstrated that slightly-robust networks learn features that can be exploited to construct highly transferable targeted adversarial examples. These adversarial examples, constructed with convolutional networks (ResNets), can attack other convolutional networks with state-of-the-art performance, as well as networks with substantially different architectures, such as the transformer-based networks including ViT, LeViT, and CCT, and with different learning objective, such as CLIP. In fact, this work is the first we are aware of that constructs targeted transferable attacks against transformer-based networks. We propose that the class-targeted transferability of adversarial examples generated with slightly-robust networks can be explained by the analogous representation-transferability of the networks. We find this to be true by showing that representation-targeted adversarial attacks generated with slightly-robust networks are highly transferable. Furthermore, since most previous transferable adversarial generation techniques rely on optimizing adversarial examples over a non-robust source network, our technique can be combined with virtually any previously existing optimization technique by replacing the non-robust source network with a slightly-robust network.

As discussed, our paper is important to the field of adversarial machine learning, as we improve transfer attacks and study the mechanism of adversarial examples. More generally, our paper reveals a phenomenon that is significant for the broader field of deep learning: we find that different non-robust networks, even when trained with similar convolutional architectures, do not necessarily have many features that substantially overlap. This can have important implications for the reliability of neural networks; when different networks rely on different features, they are susceptible to different types of errors. In addition, we present an argument that, for a given task, there are features that are useful to every tested neural network, and that these features can be learned with small-$\varepsilon$ adversarial training, even when the source network architecture and learning objective are dissimilar to those of the destination network. Thus, by studying the features of a single slightly-robust network, we can empirically discover properties that will be applicable across all non-robust networks. We speculate that this phenomenon can explain why slightly-robust networks are successful at transfer-learning tasks [43, 62, 76, 80]. With applications across the field of machine learning, we expect that the contributions in this paper will provide an important stepping stone toward discovering a general understanding of the features learned by neural networks.

Of course, our research has potential negative implications. Firstly, we propose a method to improve the generation of targeted transferable adversarial examples. While we hope that our research leads to the development of more robust and interpretable machine learning systems, in principle, an adversary could use our technique to attack existing systems. Secondly, we advocate for the use of adversarial training, which can be computationally intensive and could lead to excessive energy consumption.

## Acknowledgments and Disclosure of Funding

The authors would like to thank Rory Soiffer, Juston Moore, and Hadyn Jones for their helpful discussions and comments.

Research presented in this article was supported by the Laboratory Directed Research and Development program of Los Alamos National Laboratory under project number 20210043DR.

Melanie Mitchell's contributions were supported by the National Science Foundation under Grant No. 2020103. Any opinions, findings, and conclusions or recommendations expressed in this material are those of the author and do not necessarily reflect the views of the National Science Foundation.

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
