# A    Implementation Details

**Constructing Robust Source Networks.** We construct robust source networks by performing adversarial training [21, 45]. We use projected gradient descent in order to find model parameters $\theta^*$ that minimize the following expression:

$$\theta^* = \arg\min_\theta \mathbb{E}_{(x,y)\in\mathcal{D}}[\max_{\|\delta\|_2 \leq \varepsilon} \mathcal{L}(\theta, x + \delta, y)],$$

where $L(\theta, x, y)$ represents the cross-entropy loss of a network with parameters $\theta$ evaluated on input $x$ with label $y$. We subject the adversarial examples constructed in the inner optimization procedure to an $\ell_2$ norm constraint. We will call this constraint $\varepsilon$ the *robustness parameter* of a classifier, as it represents the ($\ell_2$) magnitude of the adversarial examples with respect to which the classifier is trained to be robust. Due to the high computational cost of adversarial training, we rely on pre-trained robust ResNet50 models that have been pre-trained on ImageNet [61]. For our experiments, we test classifiers with robustness parameters $\varepsilon \in \{0, 0.01, 0.03, 0.05, 0.1, 0.25, 0.5, 1, 3, 5\}$.

Prior research has identified a number of methods for optimizing adversarial examples given a white-box classifier $f$, many of which are based on the Iterative Fast Gradient Sign Method (I-FGSM) [21, 40], in which a perturbation $\delta_i$ is iteratively updated to maximize the loss of the network while obeying an $\ell_\infty$ norm constraint:

$$\delta_{i+1} = \delta_i + \alpha \cdot \text{sign} \, \nabla_{\delta_i} L(x + \delta_i),$$

where $L$ represents the adversarial loss function, $\alpha$ is a tunable step-size parameter, and $x + \delta_n$ is the final adversarial example after $n$ steps. At each step, $\delta_i$ is clipped such that $\|\delta\|_\infty \leq \varepsilon$ and $x + \delta_i$ is a valid image.

**TMDI-FGSM.** We adopt the state-of-the-art method recently proposed by [91], which combines three variants of I-FGSM and optimizes over many steps:

1. Diverse Input Iterative Fast Gradient Sign Method (DI$^2$-FGSM), which applies a random affine transformation to the input at each step prior to computing the gradient [87],
2. Translation-Invariant Iterative Fast Gradient Sign Method (TI-FGSM), which convolves the gradient with a Gaussian filter [14],
3. Momentum Iterative Fast Gradient Sign Method (MI-FGSM), in which a momentum term is added to the gradient [13].

Together called TMDI-FGSM, this optimization method can be described by the following process:

$$g_{\text{DI}}^{(i)} = \nabla_{\delta_i} L(T_i(x + \delta_i)) \qquad\qquad \text{(DI}^2\text{-FGSM)}$$

$$g_{\text{TDI}}^{(i)} = \mathcal{N} * g_{\text{DI}}^{(i)} \qquad\qquad \text{(TI-FGSM)}$$

$$g_{\text{TMDI}}^{(i)} = \mu \cdot g_{\text{TMDI}}^{(i-1)} + \frac{g_{\text{DI}}^{(i)}}{\|g_{\text{DI}}^{(i)}\|_1} \qquad\qquad \text{(MI-FGSM)}$$

$$\delta_{i+1} = \delta_i + \alpha \cdot \text{sign} \, g_{\text{TMDI}}^{(i)}$$

where $L$ again represents the adversarial loss function, $T_i$ represents a random affine transformation, $\mathcal{N}$ represents a Gaussian convolutional filter, $\mu$ is a tunable momentum parameter, and $\alpha$ is a tunable step-size parameter, and $x + \delta_n$ is the final adversarial example over $n$ steps.

For the DI$^2$ component of the optimization algorithm, we use a random resize and crop operation where each image is resized by a factor selected uniformly between $3/4$ and $4/3$, and then cropped to be $224 \times 224$ pixels randomly, with 0-valued padding where appropriate. Then, a random horizontal flip is applied. This is equivalent to the PyTorch code:

```
transforms.Compose([
    transforms.RandomResizedCrop(size=[224, 224],
                                 scale=(3/4, 4/3),
                                 ratio=(1., 1.)),
    transforms.RandomHorizontalFlip()
])
```

For the TI component, we apply a Gaussian filter to the gradient at each step, with the filter size of $5 \times 5$, and the standard deviation of the filter $1$.

For the MI component, we use a momentum of $0.9$.

For generating representation-targeted adversarial examples, we exclude the TI step, as we found that the representation-targeted adversarial examples were less transferable when it was included.

**Model Details.** We use a number of models for our experiments. For all robust networks trained on ImageNet, we use the pre-trained weights that are available on the GitHub page associated with Salman et al. [62]. For all convolutional destination models, we use pre-trained weights that are included with Keras [10]. For the ViT model trained on ImageNet, we use pre-trained weights from Melas-Kyriazi [48]. For the CLIP model, we use the code and weights associated with [57].

We train robust CIFAR-10 models with the Robustness library [17]. We train for 100 epochs using a batch size of 128. We include data augmentation. We optimize using standard stochastic gradient descent with momentum, using a learning rate of 0.01 and a momentum parameter of 0.9, as well as a weight decay of 0.0001. For adversarial training, we generate each adversarial example with 7 steps, using a step-size of $0.3 \times \varepsilon$ for the given robustness parameter of $\varepsilon$. For the ViT model trained on CIFAR-10, we use pre-trained weights associated with Dosovitskiy et al. [16] and finetune on CIFAR-10 for 10 epochs.

All convolutional destination CIFAR-10 models were finetuned for 20 epochs from the pre-trained ImageNet weights that are included with Keras [10].

## B    Extended ImageNet Data

In this section, we present extended data from the ImageNet.

**Untargeted Adversarial Examples.** We use the 1000 transferable adversarial examples generated to transfer to ImageNet classifiers and plot the transfer success rate when we treat the adversarial examples as untargeted, i.e., we consider every adversarial example which is misclassified by the destination classifier as a success (Figure 4). In addition, we include analogous results for adversarially trained models (Figure 5).

**Additional Tested Source-Network Robustness Parameters.** In the main paper, we exclude certain values of $\varepsilon$ in the figures that illustrate the transferability of adversarial examples for clarity, so that the results from slightly-robust networks could be more easily seen. We include the extended results for both targeted and untargeted adversarial examples (Figures 6 and 7). We observe a decrease in transfer performance as robustness increases past the optimal point. We speculate that this arises from the fact that as robustness increases, smaller features on which non-robust neural networks rely are gradually thrown away, thus reducing transfer performance.

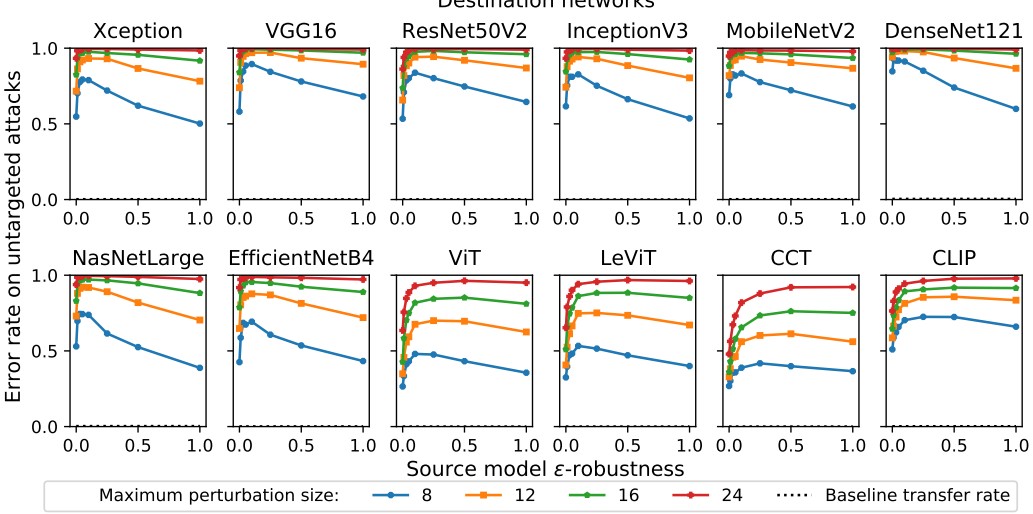

Figure 4: Error rate of destination networks (ImageNet classifiers) evaluated on untargeted transferable adversarial examples using $\varepsilon$-robust ResNet50 source models with perturbation size $\|\delta\|_\infty \leq {}^{16}/_{256}$. Higher is a more successful attack. Baseline refers to the misclassification rate of unperturbed images. (Best viewed in color.)

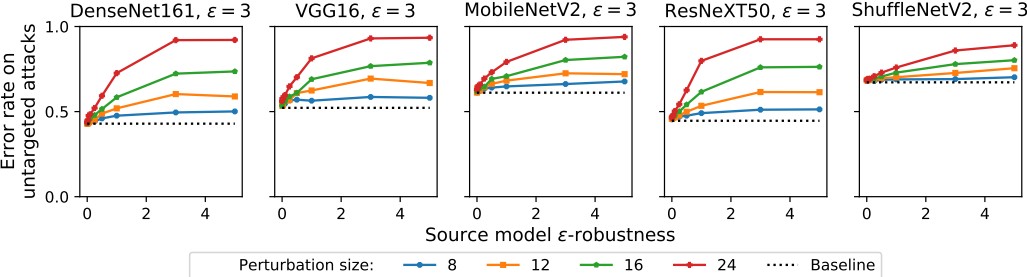

Figure 5: Error rate of destination ($\varepsilon = 3$)-robust ImageNet classifiers evaluated on untargeted adversarial examples using $\varepsilon$-robust ResNet50 source networks with perturbation size $\|\delta\|_\infty \leq {}^{16}/_{256}$. Higher is a more successful attack. Baseline refers to the misclassification rate of unperturbed images. (Best viewed in color.)

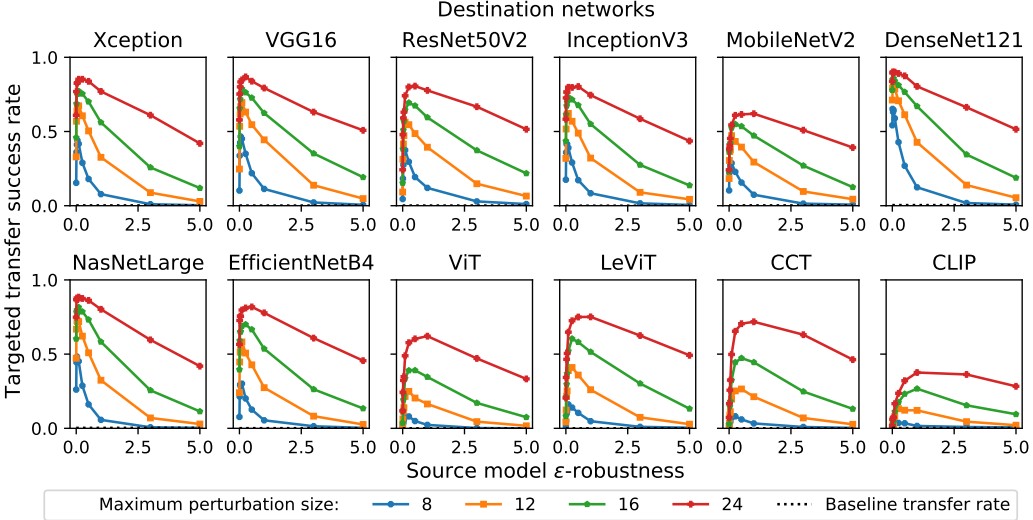

Figure 6: Extended ImageNet data (note extended horizontal axis in comparison with Figure 1): Transfer success rate of destination networks (CIFAR-10 classifiers) evaluated on targeted transferable adversarial examples using $\varepsilon$-robust ResNet50 source models with perturbation size $\|\delta\|_\infty \leq {}^{16}/_{256}$. Higher is a more successful attack. Baseline refers to the transfer rate of unperturbed images. (Best viewed in color.)

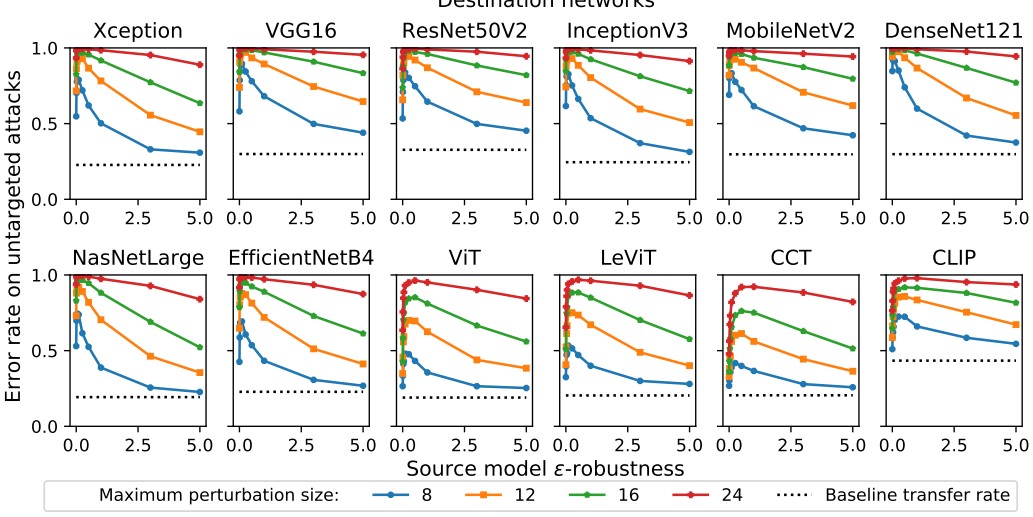

Figure 7: Extended ImageNet data (note extended horizontal axis in comparison with Figure 4). Error rate of destination networks (CIFAR-10 classifiers) evaluated on untargeted transferable adversarial examples using $\varepsilon$-robust ResNet50 source models with perturbation size $\|\delta\|_\infty \leq {}^{16}/_{256}$. Higher is a more successful attack. Baseline refers to the misclassification rate of unperturbed images. (Best viewed in color.)

# C CIFAR-10 Data

We extend our experiments to the CIFAR-10 dataset to confirm that our results are general. We present the effectiveness of targeted and untargeted transferable adversarial examples (Figures 8 and 9). In addition, we present t-SNE plots of the destination-network representations of representation-targeted examples (Figure 10), as well as the cosine similarity between feature representations and the target images (Table 3). For all experiments, our results are not as exaggerated as with the ImageNet data, but nonetheless, we observe an increase in transferability of both class-targeted, untargeted, and representation-targeted adversarial examples when we use slightly-robust source networks, confirming that our claims generalize to networks trained on the CIFAR-10 dataset.

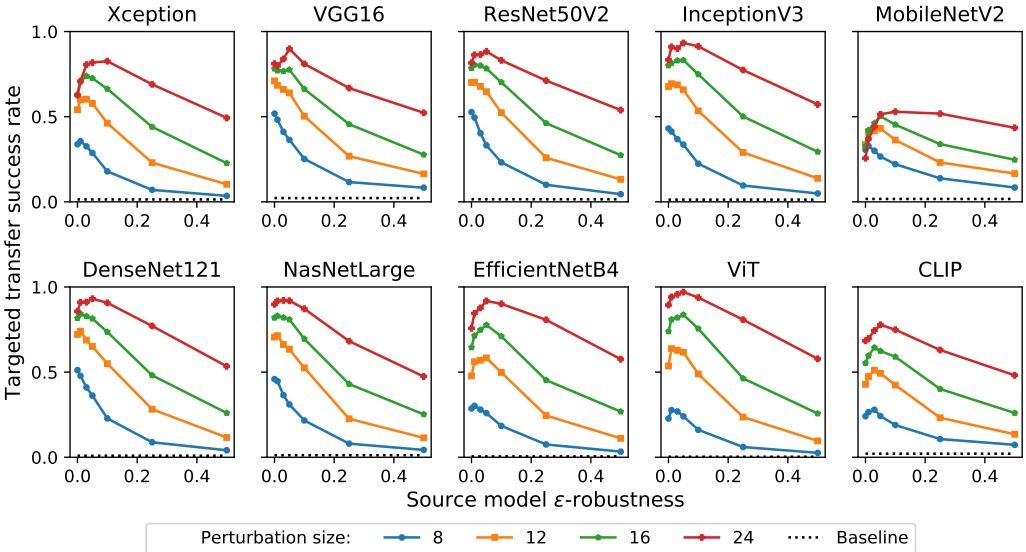

Figure 8: CIFAR-10 data: Transfer success rate of destination networks (CIFAR-10 classifiers) evaluated on targeted transferable adversarial examples using $\varepsilon$-robust ResNet50 source models with perturbation size $\|\delta\|_\infty \leq {}^{16}/{}_{256}$. Higher is a more successful attack. Baseline refers to the transfer rate of unperturbed images. (Best viewed in color.)

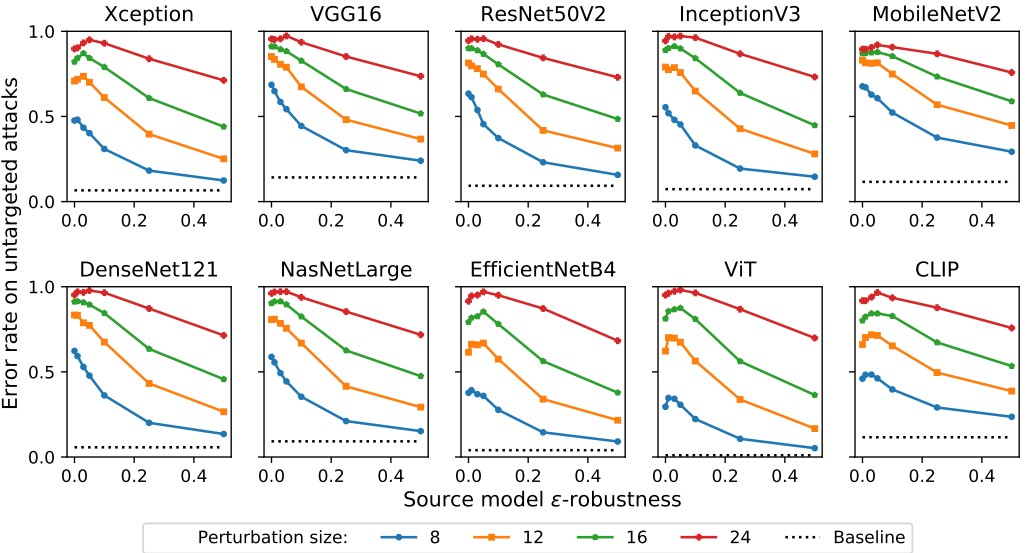

Figure 9: CIFAR-10 data: Error rate of destination networks (CIFAR-10 classifiers) evaluated on untargeted transferable adversarial examples using $\varepsilon$-robust ResNet50 source models with perturbation size $\|\delta\|_\infty \leq {}^{16}/_{256}$. Higher is a more successful attack. Baseline refers to the misclassification rate of unperturbed images. (Best viewed in color.)

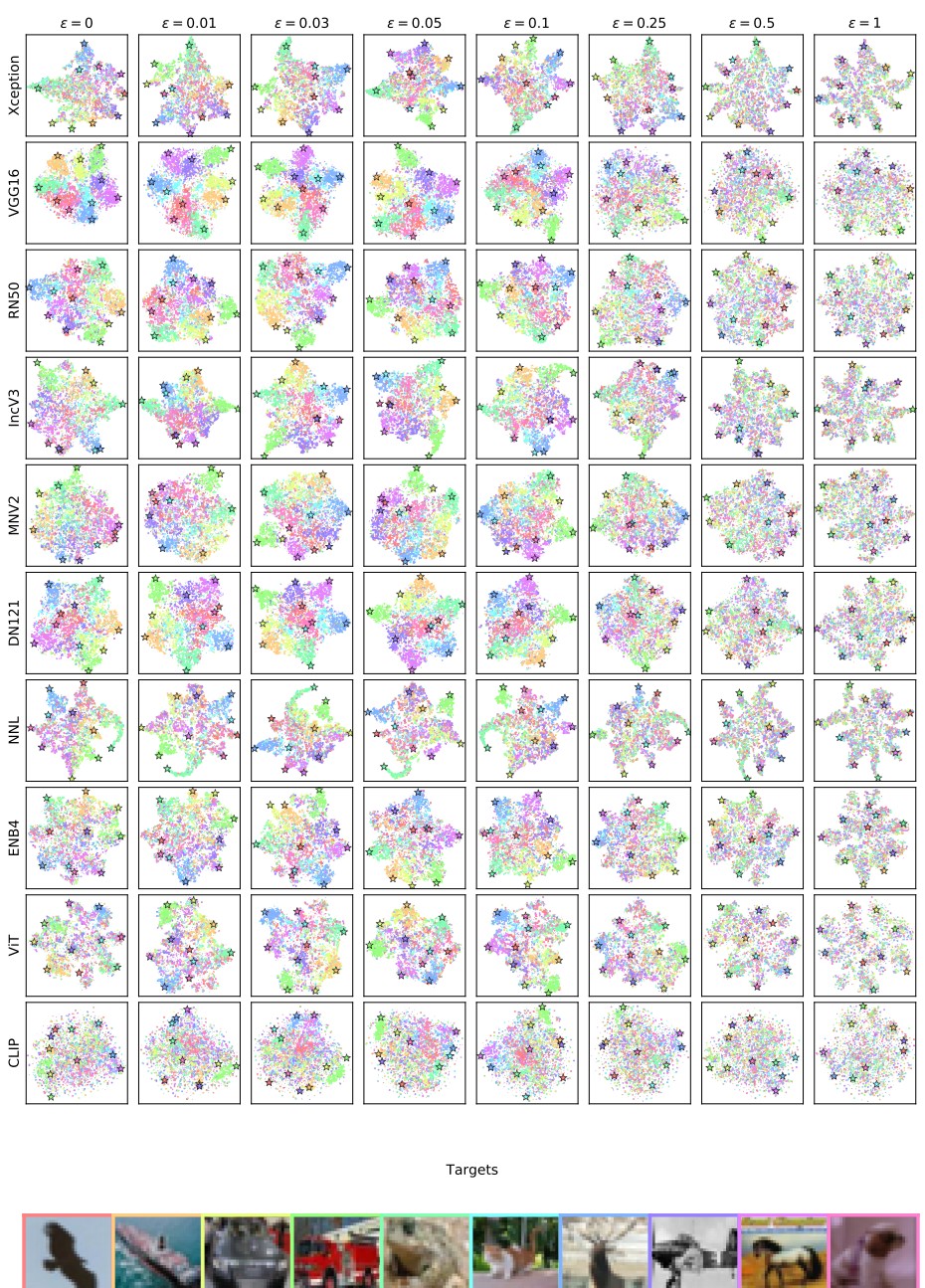

Figure 10: CIFAR-10 data: t-SNE plots of destination-network representations of representation-targeted adversarial examples generated by using whitebox ResNet50 models of specified $\varepsilon$-robustness. (Best viewed in color and magnified.)

Table 3: CIFAR-10 data: Cosine similarity between feature representations of representation-targeted adversarial examples and the targeted original images by robustness parameter of source model.

|          | 0     | 0.01  | 0.03      | 0.05      | 0.1   | 0.25  | 0.5   | 1     |
|----------|-------|-------|-----------|-----------|-------|-------|-------|-------|
| Xception | 0.609 | 0.633 | 0.665     | **0.667** | 0.655 | 0.608 | 0.552 | 0.494 |
| VGG16    | 0.736 | 0.744 | **0.749** | 0.749     | 0.728 | 0.697 | 0.668 | 0.638 |
| RN50     | 0.691 | 0.709 | **0.717** | 0.715     | 0.705 | 0.652 | 0.600 | 0.546 |
| IncV3    | 0.662 | 0.686 | **0.706** | 0.700     | 0.695 | 0.637 | 0.590 | 0.532 |
| MNV2     | 0.630 | 0.647 | 0.664     | **0.670** | 0.667 | 0.644 | 0.615 | 0.563 |
| DN121    | 0.714 | 0.726 | **0.739** | 0.739     | 0.727 | 0.683 | 0.639 | 0.595 |
| NNL      | 0.653 | 0.682 | **0.714** | 0.694     | 0.686 | 0.627 | 0.581 | 0.535 |
| ENB4     | 0.483 | 0.509 | 0.545     | **0.548** | 0.536 | 0.484 | 0.424 | 0.353 |
| ViT      | 0.269 | 0.324 | **0.375** | 0.370     | 0.362 | 0.295 | 0.211 | 0.134 |
| CLIP     | 0.768 | 0.771 | **0.773** | 0.772     | 0.772 | 0.767 | 0.762 | 0.755 |

# D   Examples of Adversarial Examples

We include class- and representation-targeted adversarial examples that have a perturbation generated with TMDI-FGSM and an $\ell_\infty$ constraint of $16/255$ (Figures 11 and 12).

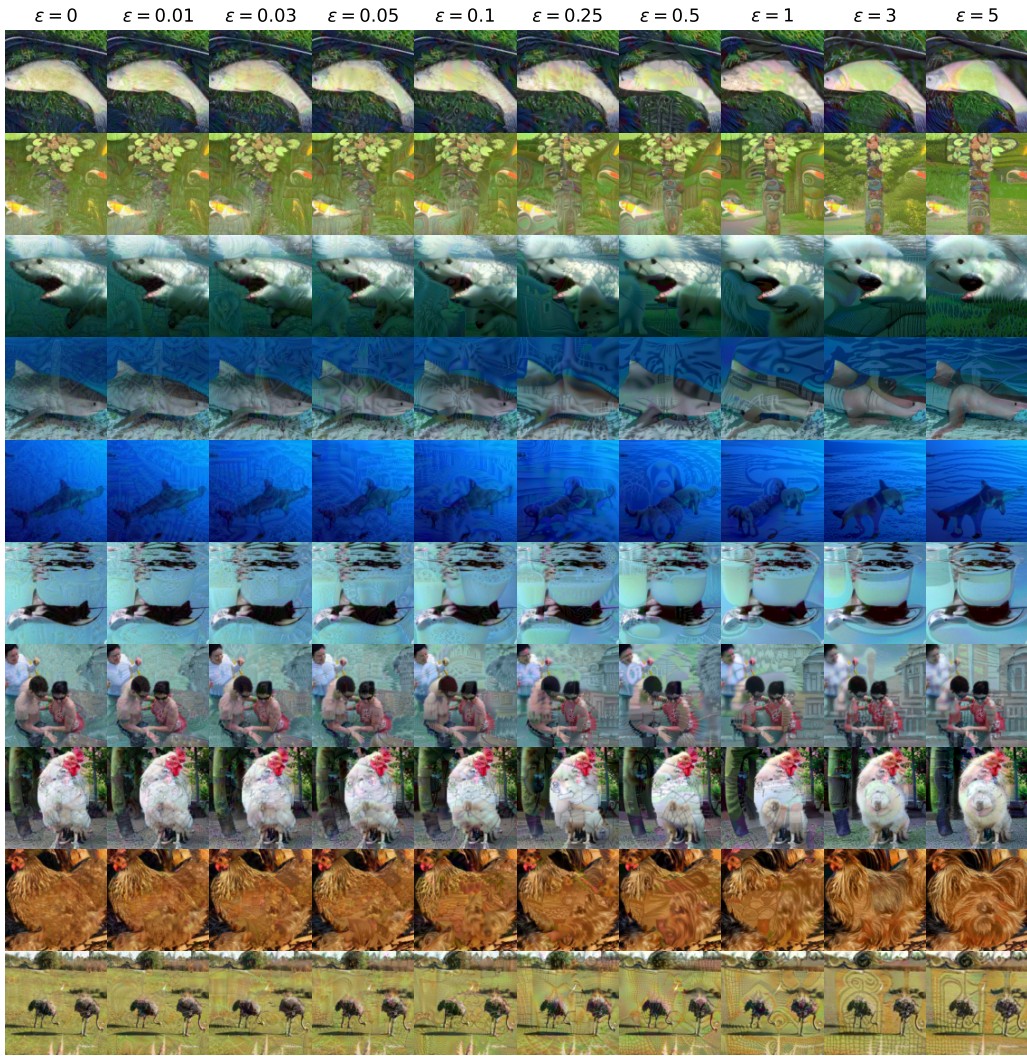

Figure 11: Examples of class-targeted adversarial examples, where the horizontal axis represents the robustness of the source network used to generate the adversarial examples. The adversarial perturbations are subject to an $\ell_\infty$ constraint of $16/255$, and are optimized with the TMDI-FGSM algorithm. (Best viewed in color.)

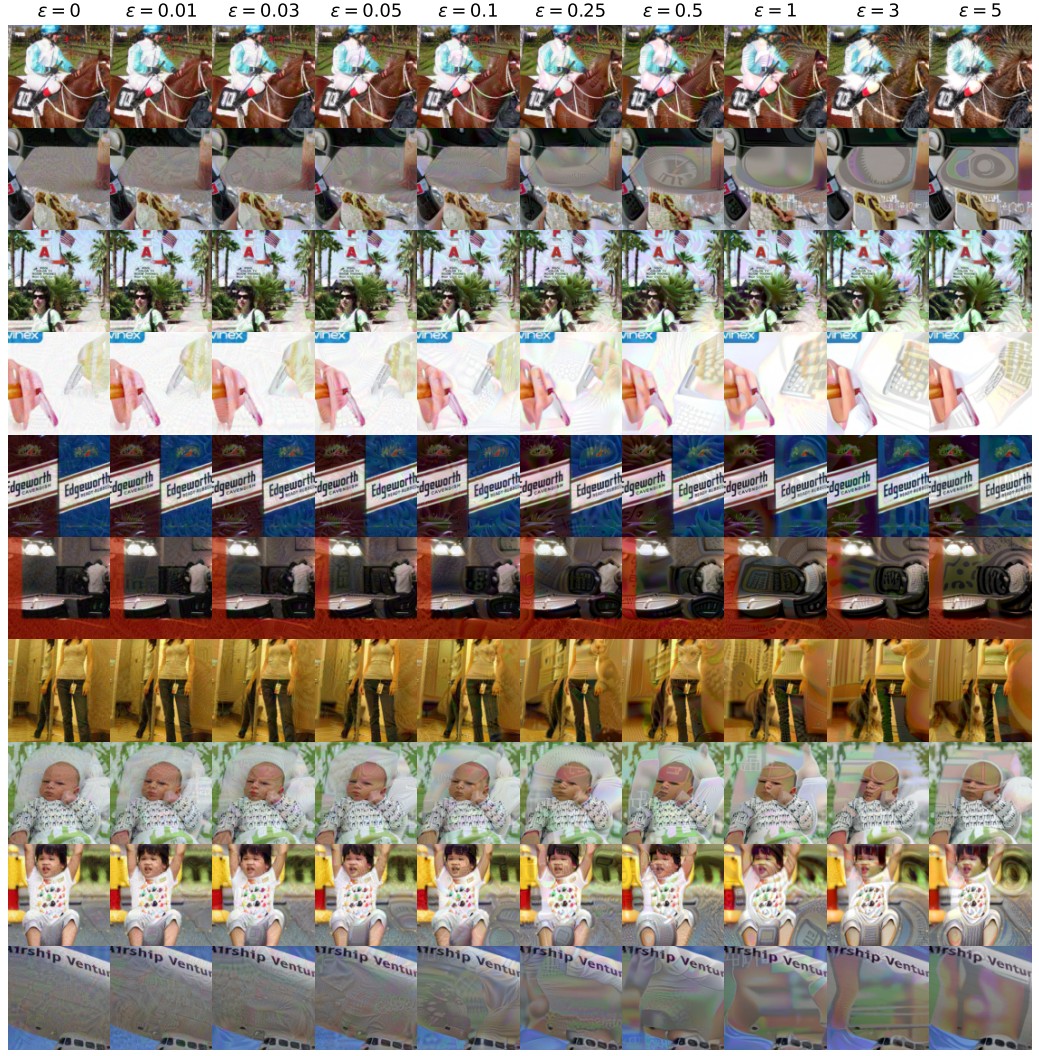

Figure 12: Examples of representation-targeted adversarial examples, where the horizontal axis represents the robustness of the source network used to generate the adversarial examples. The adversarial perturbations are subject to an $\ell_\infty$ constraint of $16/255$, and are optimized with the TMDI-FGSM algorithm. (Best viewed in color.)