# OpenReview forum: "A Little Robustness Goes a Long Way: Leveraging Robust Features for Targeted Transfer Attacks"
_NeurIPS.cc/2021/Conference — NeurIPS 2021 Poster_

### Official Review · Reviewer_ZTym · 2021-07-10

**Rating:** 7
**Confidence:** 3

**Summary:**

This paper proposes that slightly robust neural networks contain universal features that are transferable across different classifiers trained on the same dataset. The authors demonstrate empirically that adversarial examples of slightly robust networks can transfer across network architectures relative to other levels of robustness. Moreover, the authors empirically show the universality of features in slightly robust neural networks.

**Limitations And Societal Impact:**

The authors adequately describe the limitations of the work in section 6.

**Main Review:**

**Originality**
The main idea of transferability in slightly robust neural networks appears mostly novel: as noted by the authors, [67] proposes the idea that slightly robust networks are transferable. However, [67] primarily focuses on showing the entanglement of non-robust and robust features- by contrast this work focuses primarily on transferability of slightly robust neural networks. The extensive experiments provide new insights and evidence beyond what is reported in previous literature. Specifically, the experiments reveal that the internal representations of slightly robust networks are more transferable, while prior works only considered transferability at the classifier output.

**Quality**
Experiments are generally extensive and convincing. However, one question that is inadequately addressed is what the significance of targeted vs untargeted attacks is with respect to transferability. Can the authors provide an explanation for why targeted attacks are more difficult to transfer?

**Clarity**
The paper is generally clearly written and well organized. However, some of the links with prior works are not made very clear. In particular, the paper discusses transferability of non-robust features, which is a concept also discussed in [30], for example. The authors may want to further discuss the nature of features in slightly robust networks in terms of being robust vs non-robust features, and highlight how their contributions differ from prior findings.


**Significance**
If the authors can provide answers to some unanswered questions as noted above, this work could provide significant insight into the nature of transferable features in neural networks. This may lead to the development of better adversarial attacks and defenses.

**Time Spent Reviewing:**

2

---

> ### Author Response · Authors · 2021-08-10
> **Thank you for your review**
>
> We want to thank you for your thoughtful review.
>
> First we wanted to address why targeted attacks are typically less transferable than untargeted attacks. Since untargeted attacks aim to fool a network into choosing any incorrect label, and targeted attacks aim to fool a network into choosing a specific incorrect label, a successful targeted attack is also a successful untargeted attack, whereas not all untargeted attacks are successful targeted attacks. Thus targeted attacks are at least as difficult as untargeted ones, and in practice are often more difficult, i.e., state-of-the-art methods often achieve lower transfer rates on targeted attacks (compare Figure 1 and the untargeted version in our appendix).
>
> The vulnerability of networks to both targeted and untargeted attacks has huge significance for security purposes, and thus understanding how to construct highly effective attacks can help motivate defenses. However, we believe that targeted attacks are especially important for understanding neural network classifiers, as they provide a tool to compare the features of two models. When a targeted attack transfers from one network to another, it suggests that the two networks rely on the similar information for classification, and that they use the information in the same way. In our work, we show that each individual slightly-robust neural networks transfers features effectively to all tested non-robust networks, suggesting that surprising result that slightly-robust networks rely on features that overlap with every non-robust network, even though it is not the case that any particular non-robust network has features that substantially overlap with all other non-robust networks.
>
> In our revised version, we will clarify our paper’s relationship to prior findings. In particular, [30] shows that the transfer rate of untargeted attacks from ResNet50 to other networks is correlated with the architecture’s ability to learn the non-robust features that ResNet50 learns. This is consistent with, but different from, our work, as our primary contribution is identifying that slightly-robust features overlap substantially with the (non-robust) features of every tested non-robust network (which leads to transferability), whereas [30] argues that features that overlap with non-robust features lead to transferability without suggesting how one might construct highly overlapping features.
>
> Thank you again for your suggestions. If accepted, our revised version will discuss all of this in detail.

---

> > ### Comment · Reviewer_ZTym · 2021-08-29
> > **Thank you for your detailed response!**
> >
> > The authors address my all my concerns. Thus, I am increasing my rating.

---

### Official Review · Reviewer_gLyV · 2021-07-12

**Rating:** 5
**Confidence:** 2

**Summary:**

The paper demonstrates that strong transferability across models can be achieved by using slightly-robust source networks through a comprehensive study and few analyses.


**Ethics Review Area:**

["I don’t know"]

**Limitations And Societal Impact:**

Yes

**Main Review:**


- Originality:

  * This work seems to be a follow up study of [67] with a more comprehensive study, including analysis on CNN and transformer architectures, different hyperparameters, and representation similarity analysis etc. The novelty seems a bit limited.

- Clarity:
  * I found the paper a bit hard to read.
    * There are several terms that are not precise or confusing to me, e.g., transferability, universal, $\epsilon$-robustness, slightly-robust, and overlap. I found it a bit confusing in the context of adversarial robustness as it could be unclear whether those terms are for attackers or defenders.
    * There are also several typos: e.g., L157 fined -> find. L266 testset -> test set.
    * I also found it confusing that there are several sentences using future tense. It is not clear to me when it is going to happen.
    * I also found pages 7--8 a bit hard to read as there are not enough structures  but with many long paragraphs.


- Quality:
  * The paper looks at 16/255 perturbation for imagenet models. As shown in [1], 16/255 is a very large perturbation radius and it could potentially yield visible class changes.
  * I am not sure about the previous literature on transfer attack. I found it a bit suspicious to rely on FGSM type of attacks only. I wonder whether autoattack or other pgd types of methods [2] could more reliably justify the robustness of the transfer models.

- Significance:
  * I found the results and findings are interesting to know but it is mostly empirical. There is not enough explanation on why this phenomena happens, how this could be useful for defending transfer attacks, etc. The conclusion seems to expand too broad.

[1] Daniel Zoran, et al.; Towards Robust Image Classification Using Sequential Attention Models, CVPR 2020, pp. 9483-9492
[2] https://arxiv.org/abs/1910.09338

**Time Spent Reviewing:**

2

---

> ### Author Response · Authors · 2021-08-10
> **Thank you for your review**
>
> We want to thank you for your thoughtful review and we hope to address some of your concerns with this comment.
>
> In the caption for Figure 1, we say that we only use a perturbation L_infinity norm of 16/255, but this is a typo! We actually measure the targeted transferability of perturbations with L_infinity norm of 8, 12, 16, and 24 (these are the different colored lines in Figure 1). We will fix this.  For Figure 3, in our revised version, we plan to include results from different size perturbations, which follow the trend of Figure 3.
>
> We were confused about your comment about relying on FGSM types of attacks to justify the robustness of the source models. We do not verify that the source models are actually robust, but we agree that the term “robust” might be a misnomer, instead we should have used “adversarially-trained”.  Our results still hold even if our models are not fully robust, since we are using them to generate adversarial examples and not actually attacking them. However, we use the models from https://github.com/microsoft/robust-models-transfer which includes robust accuracy statistics for each model.
>
> Thank you for raising concerns and pointing out specific changes for us to make to improve clarity. These are very helpful for us and, if accepted, we will address all of your points in the camera-ready version. As for definitions, we plan to clarify all of them in the paper. Here are the brief definitions:
> - “Transferability” refers to the degree to which adversarial examples generated to fool one (“source”) network will also fool another (“destination”) network.
> - Regarding “universal”, please see our comments to reviewer QJSC. This term has caused some confusion and we believe that our results are substantial and can be discussed without introducing this term, so we plan to remove it.
> - “Epsilon-robustness” refers to the L2 norm of adversarial examples that are trained on during adversarial training. An epsilon-robust classifier should be robust to adversarial examples with L2 norm less than epsilon.
> - “Slightly-robust” is an informal term that we use to refer to models that are robust to small-epsilon adversarial examples, i.e. epsilon-robust for small epsilon (the exact range is not defined, we use it informally to refer to the epsilons where transferability is maximized)
> - When features from two networks “overlap”, we mean that they look for the same patterns in the dataset. We can measure this by measuring the degree to which adversarial examples that manipulate the features of the source network also transfer to the destination network

---

### Official Review · Reviewer_YRoB · 2021-07-13

**Rating:** 6
**Confidence:** 4

**Summary:**

This paper studies the targeted adversarial examples. They show that making the source classifier just a little robust to small-magnitude adversarial examples can improve the transferability of targeted attacks even between CNNs and transformers.

**Limitations And Societal Impact:**

The authors discussed the limitation and potential negative societal impact in Section 6.

**Main Review:**

## Originality

Though there exists a plethora of papers studying the transfer attacks, this paper provides a rather interesting perspective. The observation that making the source classifier just a little robust to small-magnitude adversarial examples can improve the transferability of targeted attacks even between CNNs and transformers is quite novel. This is motivated by the previous work (https://arxiv.org/abs/2102.05110) showing that slightly-robust features exhibit the universality principle and that slightly-robust networks can be used to increase transferability. This NeurIPS submission further provides a more comprehensive study investigating the transferability across a variety of architectures.

In summary, the originality is not about the idea of using slightly robust features, but the comprehensive study and experiments, which provides a rather solid and novel perspective on more effective targeted transfer attacks.

## Quality

The hypothesis that making the source classifier just a little robust to small-magnitude adversarial examples can improve the transferability of targeted attacks is reasonable and the empirical (both qualitative and quantitative) experiments support it.

I quite enjoy Section 4 which tries to explain why class-targeted adversarial examples transfer via the lens of universality. They measure the similarity between the representations produced by different networks.

However, I'm not fully convinced the conclusion regarding transformer classifiers is correct, as the authors only tried one particular transformer, ViT. One example is https://arxiv.org/abs/2104.05704, and the code can be found at https://github.com/lucidrains/vit-pytorch. Also, it might be more informative if they can provide the number of parameters for each model in Table 1.

## Clarity

The paper is well-written and very easy to follow.

One suggestion: the authors can indicate in the main submission that TMDI-FGSM is described in the appendix.

## Significance

Although the universality principle has been introduced and utilized for training slightly-robust networks before, this submission provides more comprehensive experiments and analysis showing that they can be used for effective targeted transfer attacks.  This is a solid contribution to the study of adversarial attacks and many researchers interested in adversarial examples will find it insightful and useful. However, the tarted audience is also thus limited.

**Time Spent Reviewing:**

1

---

> ### Author Response · Authors · 2021-08-10
> **Thank you for your review**
>
> We want to thank you for your thoughtful review.
>
> If accepted, our revised version will include the number of parameters in each model in Table 1.  We will also indicate in the main submission that TMDI-FGSM is described in the appendix.
>
> To verify that our attack transfers to other types of ViTs, we tested the attack against two more models (with available pre-trained weights for us to download), from the GitHub link you sent. The results are consistent with our claims. We can’t upload the results for you here in image format, but instead we provide a table with the results (analogous to Figure 1, in table form):
>
> - Top row: Epsilon robustness of source network (ResNet-50)
> - First column: L_infinity norm of perturbation
> - Everything else: Targeted transfer success rate
> - A higher number corresponds with a stronger attack.
>
> LeViT-256 (https://github.com/facebookresearch/LeViT)
>
> | | 0 | 0.01 | 0.03 | 0.05 | 0.1 | 0.25 | 0.5 | 1 | 3 | 5 |
> |----|----|----|----|----|----|----|----|----|----|----|
> | 8 | 0.01 | 0.051 | 0.099 | 0.115 | 0.161 | 0.14 | 0.104 | 0.048 | 0.01 | 0.004 |
> | 12 | 0.043 | 0.123 | 0.215 | 0.252 | 0.39 | 0.41 | 0.359 | 0.26 | 0.074 | 0.028 |
> | 16 | 0.082 | 0.212 | 0.297 | 0.368 | 0.521 | 0.604 | 0.581 | 0.515 | 0.302 | 0.132 |
> | 24 | 0.205 | 0.343 | 0.465 | 0.505 | 0.648 | 0.725 | 0.75 | 0.751 | 0.625 | 0.492 |
>
>
> CCT-14t/7x2 (https://github.com/SHI-Labs/Compact-Transformers)
>
> |  | 0 | 0.01 | 0.03 | 0.05 | 0.1 | 0.25 | 0.5 | 1 | 3 | 5 |
> |----|----|----|----|----|----|----|----|----|----|----|
> | 8 | 0.032 | 0.087 | 0.141 | 0.151 | 0.178 | 0.149 | 0.087 | 0.047 | 0.008 | 0.002 |
> | 12 | 0.073 | 0.188 | 0.265 | 0.278 | 0.407 | 0.418 | 0.361 | 0.253 | 0.059 | 0.023 |
> | 16 | 0.139 | 0.271 | 0.371 | 0.401 | 0.539 | 0.606 | 0.575 | 0.497 | 0.24 | 0.103 |
> | 24 | 0.221 | 0.402 | 0.492 | 0.514 | 0.666 | 0.738 | 0.755 | 0.73 | 0.615 | 0.45 |
>
> We observe the same trend as shown in the paper: a substantial increase in transfer performance to a peak, and then a decrease as robustness increase beyond the peak. We will add these results to the revised paper.

---

### Official Review · Reviewer_QJSC · 2021-07-15

**Rating:** 5
**Confidence:** 3

**Summary:**

This work studies how adversarial robustness of a source network can affect the transfer of adversarial examples to a different target network. The main finding is that slightly robust networks are ideal source networks; training the source networks with no adversarial perturbations, or with large perturbations, is less effective. The authors analyze why this might be the case, and propose that slightly robust networks have increased universality of representations.

**Limitations And Societal Impact:**

Yes

**Main Review:**

Pros:
- This fits the story of adversarially trained networks transferring better to downstream tasks (a different kind of transfer).
- The result that highly robust networks don't work as well is surprising and interesting
- The writing is engaging and clear
- The main experiments are well-designed

Cons:
Section 4 is circular. Consider the following snippets.
- “Universality Explains Targeted Transferability” (the main claim)
- “When a neural network has substantial representation transferability to every other network of a certain type, we say that the neural network has features that are universal with respect to these destination networks” (definition of universality)
- “We refer to the degree to which adversarial examples of a source model can analogously affect the features that are computed by the representation layer of the destination model as the representation transferability from source to destination.” (definition of representation transferability)

If you follow the logic, this is basically saying that the degree to which adversarial examples can perturb features in a bunch of destination models explains targeted transferability, which is tautological. This is nitpicking, but it does matter for clarity. Basically, it's currently unclear what section 4 adds. Indeed, Table 2 feels like the same thing as Table 1, except it shows cosine similarity instead of attack success rate.

I /think/ the authors were going for universality meaning that representations are similar, separate from adversarial transferability. If this is the case, then there are a number of ways to measure the similarity of two networks. If the authors could show that slightly robust networks have representations that are closer to all the destination networks according to an independent metric, that would give the analysis more internal validity.

Typos:
- 126: “using linear interpolation” → “using bilinear interpolation”

Overall, the main results are interesting, but the issues in section 4 need to be addressed.

**Time Spent Reviewing:**

2

---

> ### Author Response · Authors · 2021-08-10
> **Thank you for your review**
>
> We want to thank you for your thoughtful review.
>
> Your main concern is that we argue that transferability is explained by universality, which itself essentially measures how well classifiers transfer features. We completely agree that, as currently written, Section 4 is circular, and, if accepted, will address this in the camera-ready version of the paper. However, we do believe that section 4 proposes an interesting and novel claim, which, if we re-structure the writing, can be emphasized.
>
> Our plan for rewriting is to entirely remove discussion of the concept of “universality” instead focus on the following substantial results, demonstrated in Figure 3:
> 1.	Non-robust networks are surprisingly bad at transferring features. This suggests that even when the classification output is successfully transferred, the individual features are not substantially perturbed, suggesting that transferability from non-robust source networks arises from only a slight overlap in features, or the overlap of only a few highly vulnerable features.
> 2.	Adding only a _tiny_ amount of robustness (eps=0.1 or so) causes features to transfer well enough to form clusters in Figure 3, suggesting that transferability from slightly-robust source networks arises from substantial overlap of many features.
>
> We believe that these results are novel, interesting, and applicable to the adversarial ML community, and, if we present the results in this way, we will not be relying on a circular argument.
>
> We hope that this rewriting plan addresses your comments.

---

### Decision · Program_Chairs · 2021-09-27

**Decision:**

Accept (Poster)

**Comment:**

This paper observes that classifiers with a certain degree of robustness lead to more transferable (targeted) adversarial examples. The manuscript shows how features learned by robust classifiers transfer better to other classifiers. This provides some further insights (building on https://arxiv.org/abs/2102.05110) into transferability, a phenomenon which remains poorly understood in the adversarial ML community. Thus, there is merit to the work proposed here. I encourage the authors to take into account discussions from the reviews while preparing the camera ready of their manuscript, and to open-source their code. In particular, the authors acknowledged that their discussion of universality is hard to follow, if not a bit circular, so I recommend following suggestions made in the author response to rewrite Section 4. Finally, the authors conducted their experiments using a single optimizer (see line 71), I would encourage them to confirm their findings on additional optimizers to obtain a more complete result.